# Rare Diseases in the Educational Field: Knowledge and Perceptions of Spanish Teachers

**DOI:** 10.3390/ijerph19106057

**Published:** 2022-05-16

**Authors:** Ramón García-Perales, Ascensión Palomares-Ruiz, Lydia Ordóñez-García, Eduardo García-Toledano

**Affiliations:** Department of Pedagogy, Faculty of Education of Albacete, University of Castilla-La Mancha (UCLM), 02071 Albacete, Spain; ascension.palomares@uclm.es (A.P.-R.); lydiahonrubi@gmail.com (L.O.-G.); toledanoeg@gmail.com (E.G.-T.)

**Keywords:** rare disease, education, inclusion, teachers, knowledge and perception

## Abstract

Background: Education plays a fundamental role in everyone’s wellbeing. That means it is essential to provide quality inclusive activities to ensure equity and equality of opportunity in order to shape a cohesive, democratic, healthy society. Methods: In this study we focus on how inclusive educational practice addresses students with rare diseases, looking at teachers’ knowledge and opinions in this regard. A questionnaire was administered to 574 teachers who taught in various stages of non-university education to determine their knowledge and opinions about different dimensions: conceptualization, legislation, intervention, and diagnosis. Results: The results suggested various ideas for improvement in pursuit of positive, real inclusion, such as the need to improve teachers’ knowledge and understanding of these students’ characteristics and potential, with widespread specific training being urgently needed. Conclusions: in summary, students’ rights to education without discrimination is a basic premise of an educational system, leading to the need for a complete educational response that allows each student to develop as a person.

## 1. Introduction

Inclusion means making space for any person within a given activity, group, or service, which also covers the educational field [1]. An inclusive society means that we are all part of it, with equality in conditions of access and conditions to remain part of it. Striving to avoid exclusionary attitudes promotes equality, respect, and non-discrimination [2]. Everyone’s wellbeing depends on that; inclusion benefits each person who experiences it and provides an effective means of education for all, regardless of the characteristics of a given student [3].

Educational and social inclusion for all children is a fundamental premise of an egalitarian, quality education system. Equity is an additional tool for developing teaching and learning processes in response to each person’s individual and social needs as well as helping them to overcome obstacles that may hinder their learning [1]. All members of an educational community should ensure it is put into practice. The current education legislation in Spain—Organic Law 3/2020, 29 December, amending Organic Law 2/2006, 3 May, on Education [4]—states that:

The responsibility for all students’ academic success does not solely fall on the efforts of each individual student, but also on their families, teachers, schools, education authorities, and ultimately, society as a whole. In other words, to ensure quality education for all, it is essential to have the commitment of each part of the educational community and society as a whole. One of the most important consequences of the principle of shared effort is the need to deliver equitable schooling to each student [4] (p. 122869).

In the autonomous community of Castilla-La Mancha, where the researchers work as educators, and hence where this study took place, educational inclusion is defined as:

The set of actions and educational measures aimed at identifying and overcoming barriers to learning and participation for all students, promoting educational progress for all, considering different abilities; rates and styles of learning; motivations and interests; and personal, social, economic, cultural, and linguistic situations; without equating difference with inferiority, such that all students are able to make the best possible use of their potential and individual capabilities [5] (pp. 32232–32233).

There has been a long, varied progression in the education delivered to those with special educational needs to promote their inclusion. It has gone from a past characterized by discrimination and a lack of knowledge, to the implementation of educational policies and actions focused on equality of opportunity for all. School should be a space for reflection in order to facilitate a more inclusive educational environment [6]. Traditionally, in all social groups, the manifestation of certain personal capabilities has led to certain expectations [3]. Everyone forming part of any social system has some potential and some distinctive characteristics. One of the many distinctive features of the current Spanish educational system is the heterogeneity of the student body and the complexity and multidimensionality of educational activity.

Part of this student heterogeneity, and what we focus on in this study, is students who suffer from rare diseases. These are defined in Europe as diseases with a prevalence below 50 cases per 100,000 people and in the USA, below 80 cases per 100,000 [7]—they are “uncommon” (poco frequente), which is one term used in Spain. Despite this, there are many of them, 6053 rare diseases are recorded in the Orphanet database with information about prevalence and incidence [8]. They are varied, with very different profiles [9]. They have multi-system impacts and are occasionally non-visible [6], and they are mainly genetic or congenital [10]. They are often unpreventable and untreatable [7], and may be chronic, and sometimes the cause of an early death [11]. This makes them a priority in the emerging area of global healthcare [11,12], as they are considered “largely unknown at the healthcare, social, and educational level” [6] (p. 7).

Because of this, these diseases need greater visibility [13] and a multidimensional approach that covers various areas of action [14]. For example, there should be increased healthcare research resources in order to achieve more accurate, effective diagnosis and treatment [11]. In the social setting, patient support groups should be more widespread to encourage an activist, welfare-based approach to those suffering from these diseases and their families [15]. In education, inclusion (both educational and social) should be promoted by use of individualized education that addresses the distinct potential and characteristics of those suffering from these diseases [16]. The importance of extending these activities to as many areas as possible has only grown with the COVID-19 pandemic, which has had a negative impact on the quality of life for those suffering rare diseases [17] and has led to the need for education in managing COVID infections in this population [18].

Rare diseases cover a broad range of different types of illnesses, which means there needs to be investment in resources that raise their visibility and allows them to be categorized using a standardized conceptual classification [19]. One example of this that is worth highlighting is EURODIS-Rare Diseases Europe, a non-profit organization that encompasses patient advocacy groups in more than 74 countries that take action to improve the quality of life for rare disease patients and their families. Raising the visibility of these illnesses is essential through expanding research, policies, and services that respond to the needs of patients and their families [20,21].

In Spain, there are two benchmark organizations that support and promote the inclusion of those with rare diseases and those with potential diagnoses in order to improve their quality of life [The National Center for Care of Patients with Rare Diseases and their Families: Centro de Referencia Estatal de Atención a Personas con Enfermedades Raras y sus Familias (CREER) and the Spanish Federation of Rare Diseases: Federación Española de Enfermedades Raras (FEDER)]. CREER’s objectives are aimed at supporting and encouraging educational inclusion for students with rare diseases, through the following areas of intervention [6] (p. 4):

Individual and/or group attention for children and their families; information, advice, and co-ordination with counselors and tutors in schools; awareness and education programs in schools; specialized training for education of education professionals, both active and in training; and participation in collaborative projects with other organizations.

FEDER’s interventions are in the social, healthcare, research, and education fields. It espouses and attempts to put into practice its values of campaigning, hope, commitment, unity, solidarity, participation, and transparency [22]. FEDER considers education to play a fundamental role in social inclusion for students. In this educational arena, FEDER promotes various activities to drive inclusion for students during their schooling. Two examples of projects stand out for non-university students: ‘Rare diseases are already at school with Federito’ [Las Enfermedades Raras ya están en el cole con Federito] aimed at students in primary school and pre-school (aged 3–12 years old), and ‘Take on an uncommon challenge’ [Asume un reto poco frecuente], aimed at students from the 5th and 6th years of primary school, secondary and further education, and vocational training (from about 10 years old and up).

In the present study, we highlight the importance of teaching in the processes of diagnosis and educational interventions aimed at students with rare diseases [6]. Many studies have been undertaken on the role of teachers in putting inclusion into practice [23,24,25]; however, there are practically no studies that have stressed rare diseases and the importance of educational inclusion for those students from the teaching perspective. Some data that is worth considering comes from the ENSERio study [26]: 9.48% of people with rare diseases reported that the school did not give them the individual attention that they needed; 14.23% needed the support of specialist personnel that the school failed to provide; 16.46% needed technical resources that the school failed to provide; and 41.30% felt discriminated against at some point in the educational environment. There are other aspects that affect these students’ educational wellbeing and their teaching/learning processes [26]: the costs associated with coping with the disease; the extent of any recognized disability; any psychological help required; opportunity costs, and costs of access to work. Faced with this reality, we asked what would be the reality of students with rare diseases, and what dimensions and elements may affect their educational wellbeing from a teachers’ perspective?

With the present study, we sought to assess what teachers in various schools in one Spanish region know and think about rare diseases by applying a questionnaire covering various dimensions for analysis, created for this purpose. By doing so, we aim to help the expansion of research for educational inclusion of students with rare diseases.

## 2. Materials and Methods

The study used a quantitative, descriptive, ex post facto methodology to analyze the relationships in a set of numerical data.

### 2.1. Participants

The sample was selected using simple random sampling following potential subjects demonstrating willingness to participate. A total of 574 teachers working in various schools at various educational stages completed the questionnaire. They were from the five provinces that make up the autonomous community of Castilla-La Mancha: Albacete (179: 31.18%), Ciudad Real (112: 19.51%), Cuenca (137: 23.87%), Guadalajara (64: 11.15%), and Toledo (82: 14.29%). Almost a third of the teachers (170: 26.92%) were men, while 404 (70.38%), were women—this is a similar ratio of men to women who teach in schools in this region (men 29.67%, women 70.33%) [27]. The distribution of participant ages was as follows: 15 (2.61%) were aged 21–25; 122 (21.26%) were aged 26–35; 296 (51.57%) were aged 36–50; and 141 (24.56%) were over 50 years old. The participants’ professional experience was as follows: 83 (14.46%) had between 0–5 years’ experience; 165 (28.75%) had between 6–15 years, 210 (36.58%) had between 16–25 years; and 116 (20.21%) had more than 26 years of teaching experience.

### 2.2. Instrument

The questionnaire used in the study contained 20 items spread over four dimensions: conceptualization, legislation, intervention, and diagnosis. The operational definitions of the dimensions are as follows:Conceptualization: key aspects allowing understanding of rare diseases, from definitions to characteristics, prevalence, and types;Legislation: legislative frameworks influencing the educational response to these students that teachers should be aware of;Intervention: practical aspects affecting teaching and learning processes with these students in schools;Diagnosis: content related to identifying these students, warning signs, characteristics, and procedures, as well as assessment of the level of awareness about detection and research in the field.The structure of the instrument is given below (Table 1).

The scoring for each item was: Not at all (1), A little (2), Moderately (3), A fair amount (4), and Very much (5). In addition, the final part of the questionnaire was designed to collect qualitative contributions from the participants, with the instruction “In this section, we ask you to include any information you think should be added to the questionnaire that has not been covered by the items above, noting any observations or suggestions for improvement”. This information made it possible to complete the results, as can be seen in the Results section below.

The index of reliability, using Cronbach’s alpha, was 0.94 for the total item score in the questionnaire. Internal consistency, relating each dimension to the items making it up, gave the following values of Cronbach’s alpha: Conceptualization (items 1–5) 0.80, Legislation (items 6–10) 0.79, Intervention (items 11–15) 0.85, and Diagnosis (items 16–20) 0.79. Content validity was assessed by a panel of 7 experts with broad practical knowledge of educational inclusion, four of whom had specific training in rare diseases. The indices for content validity were: Conceptualization 0.77, Legislation 0.94, Intervention 0.82, and Diagnosis 0.88. Overall content validity was 0.85 with a Kappa index of 0.88. Finally, with regard to exploratory factor analysis, the KMO index was 0.94, Bartlett’s sphericity test gave a value of 7699.08 and *p* < 0.001, the percentage of total variance explained was 68.49% and the factor structure was similar to the initial four-dimension design (see Appendix A). In confirmatory factor analysis, the correlations between the items and their respective latent variables or dimensions were adequate, indicating a plausible fit for the model with CMIN = 32.48, *p* = 0.09, RMSEA = 0.05, CFI = 0.95, and TLI = 0.94, confirming the fit of the structure to the data.

### 2.3. Procedure and Data Analysis

Data was collected between November and December 2021. An introductory letter was first sent to school headteachers in Castilla-La Mancha outlining the study content and objectives, asking them to participate and giving them a link to the questionnaire for them to share with their teachers if they agreed to take part. The data was anonymous and the confidentiality of school and teacher data was ensured. Data analysis was performed using SPSS version 28. In addition to the statistical justification above, the following statistics were calculated: frequency, percentage, minimum, maximum, mean, standard deviation, asymmetry, and kurtosis.

## 3. Results

The results for the questionnaire as a whole were as follows (Table 2).

As Table 2 indicates, the mean score for the instrument—out of a possible maximum of 100—was 66.49 (SD = 16.41), with negative asymmetry and negative kurtosis (platykurtic). This is clear in the chart showing the frequencies of the scores (Figure 1).

Moving on from the overall scores, more detail is given below for each of the dimensions making up the questionnaire: conceptualization, legislation, intervention, and diagnosis. For each dimension, results are given for each item and the dimension overall, supported by the teachers’ observations and suggestions for improvement based on the qualitative analysis. Each item has a minimum of 1 and a maximum of 5, and for each dimension the minimum is 5 while the maximum is 25. In addition, a qualitative analysis was performed by considering the teachers’ contributions to the observations section of the instrument.

The results for the Conceptualization dimension are given in Table 3.

The items on which the teachers generally scored highest in the Conceptualization dimension were items 1 (I know what a rare disease is) and 4 (When I have had a student with a rare disease in my class, I knew what their main characteristics were). In contrast, the teachers generally gave lower scores to items 2 (I know the categories that the different types of rare diseases fall into), 3 (I know how prevalent the rare diseases I have dealt with in my school are at a national or international level), and 5 (We have maintained contact from the school with patient advocacy groups that deal with rare diseases). This indicates a need for additional, broader training for teachers regarding the conceptualization of rare diseases. In the questionnaire, some teachers indicated awareness of their lack of training on the topic, although they indicated that completing this questionnaire had sparked their interest in it, and their schools’ educational responses to this group. Extended training activities would probably lead to greater awareness of the individualized educational response to these students.

In addition, teachers thought it was fundamentally important to involve external services, whether strictly educational or not, in the process of improving the conceptualization of the potential and characteristics of these students. External services refers mainly to healthcare professionals and patient groups who care for these students outside of teaching hours. The co-ordination between them could be better, and teachers said that it was occasionally non-existent. These services should give schools the opportunity to understand their work and offer to work in collaboration. Teachers occasionally reported feeling “unprotected” as they did not know who to contact or how to behave in specific cases.

Teachers also noted that although these students were the minority in their classrooms, that should not mean that their education should take a back seat. Each student is unique, and their potential and characteristics should be addressed. Teachers also remarked on the breadth of diseases, sometimes very different from each other, which had a negative impact on their knowledge and generalization of potential, characteristics, modalities, and intervention approaches between one disease and another [11]. It is essential to have information on characteristics, percentage incidence, and reports of activities and successful experiences in educating these students. There are examples of documentation that meet these requirements [6]. All of this must be considered in all stages of schooling, compulsory or otherwise.

To complete this analysis, the results for central tendency, dispersion, and distribution are given below (Table 4).

These results reinforce what the frequencies already indicated, the need for better teacher training in this area. This can be seen in the total score for this dimension, at 13.85 (SD = 4.74), it is a long way from the maximum score of 25.

The results for the Legislation dimension are given in Table 5.

Table 5 shows that four of the items in the Legislation dimension were mostly scored highly by the teachers. These were items 6 (I know the educational inclusion legislation in our region), 8 (I have read and understand the article in the Spanish Constitution which describes the right to education for all), 9 (I understand the importance of there being a law aimed at educational inclusion), and 10 (Within my teaching, I believe inclusive education to be important, and therefore, the principles of inclusion must be applied in the classroom). The frequencies of scores for item 7 (I know the content of the most recent education legislation, especially about student educational needs) were more balanced. This indicates that teachers successfully deal with legislation related to students’ educational inclusion, especially the older legislation, and understand the need for inclusive normative frameworks which are put into practice with students in the classroom.

In contrast, there is a clear need for legislation to support the work undertaken by patient groups which deal with these groups of students. Teachers believe that there should be better legal and financial support for creating and expanding more groups so that the geographical reach of their interventions can be as wide as possible.

In addition, the results indicated the importance of having legally regulated protocols to follow in schools, families, healthcare, and patient groups. Making it compulsory to put certain inclusion measures into practice may drive shared responsibility in all settings surrounding students, along with greater communication and awareness of these students’ potentials and characteristics. This should be a public policy priority [11].

These protocols should include times and places for meetings between members of the educational community and other agents who may have an impact on educational processes. In this regard, peers and families of students who share classes with students with rare diseases should be included. Openness and dialogue are essential. The aim of all of this is to facilitate assistance for the student via the transmission of detailed, practical information that will be the foundation of a proper, individualized educational response. It is important to be aware of the high levels of suffering that many families of students with rare diseases experience, and it is essential to have a multidisciplinary approach from the full educational community in order to raise levels of awareness and knowledge.

The protocols should also include a catalogue of currently known rare diseases, together with their characteristics and classification. One request from teachers in this regard was an indication of the most commonly encountered diseases in education nowadays, something teachers believe would make their work easier.

The descriptive statistics for the Legislation dimension are given below (Table 6).

Table 6 clearly shows the same trend as the frequency analysis, with higher mean scores in items 6, 8, 9, and 10, and item 7 having a lower mean score than the others at 3.19 (SD = 1.47), indicating that teachers have not updated their knowledge of the most recently passed education act. The mean total score for the dimension, at 20.03 (SD = 4.10), was close to the maximum of 25, reflecting the high scores in this dimension.

The results for the Intervention dimension are given in Table 7.

The scores in the items from the intervention dimension were varied. The teachers generally gave high scores to items 12 (I have educated myself about the signs and symptoms students may exhibit if they suffer from a rare disease) and 13 (I think the family-school relationship is essential for proper intervention for students with rare diseases). They gave mostly low scores to items 11 (I have had, or still have, a student in my class with a rare disease) and 15 (I am able to advise and guide other teachers about activities with these students). Item 14 (I know specific activities to do with these students), on the other hand, had scores which were relatively balanced across the five categories.

This indicates that teachers are predisposed to educating themselves about the distinct characteristics of these students so that they can deliver individualized education. This education is mostly through the internet, and teachers are aware of the importance of family involvement in achieving this goal. Teachers exhibited positive attitudes towards inclusion [28,29]. They also indicated that families had sometimes tried to ‘hide’ the diagnosis, which had occasionally affected the students’ education.

Some teachers also exhibited difficulty in identifying when a student had a rare disease, which is consistent with what was noted in the Conceptualization dimension, and they would find it difficult to feel able to offer advice and guidance to other teachers about diseases they may have encountered previously in their careers. Some also indicated that they knew specific actions and activities due to specific cases they had in their classrooms.

In contrast, the participating teachers reported that there should be more means offered for educating these students from education authorities, indicating a need for more personnel and material resources. They indicated the importance of being able to have healthcare professionals in schools that these students attend, particularly nurses. In addition, although pediatricians and other medical specialists are not part of the school, the teachers felt that close co-operation was essential, whether occasional or frequent, depending on the nature of each student. Ultimately, this suggests the position of a health coordinator in schools, a teacher who would be responsible for co-ordination between their school and any healthcare services students need. They would need appropriate training to work effectively, along with authorization from families in order to exchange and access patient information.

Teachers were aware of their lack of information about these diseases and noted the need for training which would help improve the wellbeing at school for these students. Initial teacher training, mostly delivered at university [28,30], and continual teacher training, delivered by teacher training centers or private institutions, are essential, and a current objective considered by education administrations [4]. Teachers are aware of the need for better training in order to be able to respond to the demands of inclusive education, as other studies have shown [23,31,32,33,34].

In this relationship between training and educational intervention, the teachers underlined the importance of training in new methodologies that would allow them to deal with all their students, with particular attention to incorporating technology and using new organizational and support structures, considering the peculiarities of each school. The teachers indicated that, with respect to organization and in order to deliver suitable education, they need smaller class sizes, something that has been shown to promote inclusion. In the present study, some teachers indicated that their knowledge of rare diseases and how to approach them in education came more from their practical experience than from participation in any training activities, something which merits particular consideration.

One of the aspects teachers noted that deserves more development is the use of technology for inclusion at school [35,36,37]. There are many resources that can be used in educational processes to encourage inclusion, at the individual level, or in small or large groups. Teachers must be aware of their benefits and drawbacks [38], so that they can use them as effectively as possible in teaching–learning processes with their student groups [39].

When we think of technological resources, we should not solely consider those aimed at individual students. For example, videoconferencing platforms can be used for meetings with patient groups where the location makes it hard to have those kinds of meetings, such as rural schools which do not have such groups nearby.

All of this in relation to the Intervention dimension is essential to ensure the individual wellbeing of students, in all dimensions. The teachers exhibited concern about the integration of these students and their emotional development, key issues in emotional and behavioral adjustment [40]. Preventing feelings of being isolated, alone, or ignored must be key in inclusion for these students. The teachers also highlighted peer bullying or mistreatment that might occur [26]. Mutual enrichment between everyone in a school is possible, and all of those working in schools can help promote the educational, social, and cultural integration of all of their students [41].

The statistics for central tendency, dispersion, and distribution for the Intervention dimension are given in Table 8.

Table 8 reflects the variety of the results noted above in the frequency analysis. The mean scores for items 12 and 13 were the highest, at 3.63 (SD = 1.47) and 4.50 (SD = 0.89), respectively. The lowest mean scores were in items 11 and 15, at 2.61 (SD = 1.52) and 2.58 (SD = 1.43), respectively. The mean total score for the dimension was 16.27 (SD = 5.40), which is high but still far from the maximum possible score. Educational intervention following identification is essential, and how well the teaching/learning process fits to the characteristics and potential of each student depends on this.

Finally, the results for the Diagnosis dimension are given in Table 9.

Table 9 shows the variation of the results for each item. There were generally high scores for items 19 (I think the relationship between the school and the families is essential in order to be able to properly detect and help an initial diagnosis) and 20 (I am aware of the need for more research to help diagnosis and treatment), with the latter being scored highest in the whole questionnaire. In contrast, there were generally low scores for items 16 (I know the warning signs a student with a possible rare disease may present) and Item 18 (I know how to clinically diagnose a rare disease). Item 17 (I educate myself and try to contact other diagnosed cases in order to improve the education I give to students with rare diseases that I might encounter as a teacher) had relatively balanced scores over the five categories.

The teachers were aware of the importance of good communication between the school and the family in order to not only provide suitable educational interventions, but also to confirm a differential diagnosis. They also exhibited high awareness of the need to employ greater personal, material, and economic resources to improve detection and treatment, something that many patient groups dealing with these students have been asking for [11]. Quality of life for these students is fundamental, and this need is made more acute by the chronic nature, degeneration, or premature death associated with many of these diseases. The data about when children are diagnosed with these diseases underscores the importance of the school years [6,42]: 69.9% of rare diseases are diagnosed in the pediatric age group (up to 18 years old, depending on the region), 18.2% are diagnosed either in that age group or in adulthood, and 11.9% are diagnosed in adulthood (from the age of 18 onwards).

Teachers need more training in the warning signs and indications of a possible diagnosis, and in the importance of having knowledge about identification procedures, for which information provided by the school is often important for proper differential diagnosis. There is also a need for teachers to connect with and obtain information from already diagnosed cases so they can properly adjust their teaching to these students’ characteristics.

The participating teachers noted the importance of close relationships between the educational arena and healthcare, highlighting that, at times, late diagnoses had resulted in education that was a poor fit to students’ characteristics. Teachers need to be aware of a diagnosis as early as possible in order to fully include students in the school. They also indicated that within this close cooperation between schools and healthcare, it is important for them to have training in basic first aid when they have students whose diseases might require it. As an example, the following two links lead to pages which explain the care to provide in the case of two rare diseases [6]: West Syndrome and other epileptic encephalopathies (Available online: https://cutt.ly/PAqYcGs (accessed on 12 February 2022) and Congenital Metabolic Disorders (Available online: https://cutt.ly/KAqYDnH (accessed on 16 February 2022).

In this regard, the teachers reiterated the need for more staff and reduced waiting times for care from healthcare services, such as neuropediatric and child mental health units. They also noted the importance of these issues in early care services and local health centers. According to the teachers, these services need the resources to expand early identification and avoid excessive waiting times which make coordination difficult. Early detection will trigger preventive action and rehabilitation as quickly as possible. Some studies have highlighted the importance of reinforcing the health system so that it can properly and promptly care for students with rare diseases [10,20,43,44,45], and that they are a significant problem for public health and a challenge for medical care [46].

Infant education, between the ages of 3 and 6 in Spain, is fundamental for this early identification. Although it is not a compulsory stage of education, most of the students who are diagnosed with rare diseases are in the first year of infant education. This is why advice and guidance are key to inclusion at the beginning of the students’ schooling; how they are received is a cornerstone of their educational care [6]. First impressions can be fundamental and may help determine students’ adjustment to school and their academic pathway. It is essential to expand this early work on inclusion at these ages to subsequent phases of schooling.

Lastly, it is worth noting the importance of homeroom or form teachers. The homeroom teacher coordinates the activities of a set group of students. Because of that, it is their responsibility to know the peculiarities of their class and strive for a proper classroom climate. This means that they coordinate all the inclusion measures in a specific school and so need to know each student’s characteristics and potential. Part of their role is to act as a link to the families of students with rare diseases and in coordination with the teachers that teach their form group. Other professionals, in addition to homeroom teachers, also do important work, including guidance counsellors and support staff. They are specialists, and fundamental parts of school inclusion practices, and there should be more of them in school teams. Where necessary, together with the family, the teaching team should provide information about specific students so that all the members of the educational community are aware of and sensitive to the specific case. Greater social, community, and institutional involvement is fundamental [3].

The descriptive statistics for the Diagnosis dimension are given in Table 10.

Table 10 shows the differences between the mean scores in the items, notably the very high mean score for item 20 of 4.72 (SD = 0.71) which is close to the maximum of 5. Item 18 had a mean score of 1.97 (SD = 1.15), indicating the importance of showing teachers the vital role they can play in diagnosing a rare disease. Schools are in continual communication with parents and education professionals can provide important information to healthcare workers that might help in differential diagnosis. Diagnostic processes are fundamental in determining students’ characteristics and potential, and based on this, establishing the conditions and resources that are needed for inclusive educational processes that ensure equity and quality.

Figure 2 gives a visual, comparative summary of the frequencies in each dimension in order to compare the differences.

It is clear from the chart that the Legislation dimension had the highest scores (M = 20.03; SD = 4.10). A graphical representation of the means between the dimensions, the maximum being 25, reiterates these differences (Figure 3).

## 4. Discussion

When we talk about rare diseases, we must be aware of the broad range of potential and characteristics that the term can cover [8]. These features will have an impact on the modification and individualization of educational processes, informing diversified and multidimensional educational actions [16]. Each person is unique, and education should aim to offer quality teaching/learning processes that are equitable for each learner. Those who work in education must create an inclusive atmosphere at school and in the classroom to promote school integration [47]. Inclusion is necessary in all aspects of the education process, and schools must be able to evaluate how it is being put into practice in their educational community [48].

The mean score observed this study for all the dimensions taken together was 66.49 (SD = 16.41) out of a possible 100. Looking at the dimensions individually, the highest scores were for Legislation, reaching 20.03 (SD = 4.10), while the lowest mean score was for Conceptualization, at 13.85 (SD = 4.74). The results for the Legislation dimension show that teachers were aware of the legislation that affects the education they offer to these students, especially the less recent legislation. It is worth noting, however, that the legislation is about more generic educational inclusion, as there are no specific legislative frameworks in relation to students with rare diseases. Everyone involved in education should be aware of the importance of legislation [49] and should support the expansion of legislative frameworks to strengthen the roles of patient groups and protocols for all of the contexts in the educational response to students with rare diseases.

In contrast, the results for Conceptualization arise from weakness in teacher training in this area and point towards improvement in the communication channels with external services, mainly healthcare and patient groups, in order to broaden teachers’ understanding of these students’ characteristics and potential. To overcome this weakness, the teachers noted the importance of expanding the use of information dossiers to show characteristics, actions, and successful experiences in education. This is useful advisory material that should be in every part of the educational community in a school. One model for reference is the guide published by CREER [6].

Conclusions can also be drawn from the scores for the remaining two dimensions, Intervention and Diagnosis, with mean scores of 16.27 (SD = 5.40) and 16.33 (SD = 4.22), respectively. For the Intervention dimension, the teachers exhibited a willingness to learn the characteristics of these students and to intervene based on those characteristics, highlighting the importance of social and emotional aspects. They were also aware of the need to involve each member of the educational community, with particular consideration of the role of these students’ families and the work of healthcare professionals, along with the importance of incorporating new teaching methods into their educational processes, such as using technological resources and organizational planning and support. The difficulties in this area come from—already noted—weaknesses in conceptualization, lack of empowerment due to the diversity and types of rare diseases, which makes it hard to apply actions in one case to other cases, large class sizes, and the lack of material or human resources available for specific interventions.

For the Diagnosis dimension, the importance of good communication between school and family has already been noted with regard to helping identify rare diseases, something that is not the exclusive responsibility of the health system. The importance of interdisciplinary and multidimensional approaches has also been mentioned [16], highlighting the role of homeroom teachers and an increase in healthcare resources in order to carry out detection processes. The teachers also stressed the importance of greater research resources in order to strengthen diagnosis and treatment, and this was the item which had the highest mean score in the questionnaire. All of this will result in fewer obstacles to diagnosis [50], for example through better knowledge of possible warning signs (something which highlights the importance of continued training), and fewer difficulties in offering modified education, promoting better wellbeing and quality of life from the first stages of schooling.

## 5. Conclusions

The teachers who participated in this study gave a positive assessment of it. This is because, by completing the questionnaire, they were more aware that their knowledge and training was insufficient for providing a proper educational response to the potential and characteristics of these students. Expanding training processes will be essential in overcoming this pattern. Currently, updating knowledge through continual training has become indispensable.

The main limitation of this study was the focus on a single autonomous community in Spain, Castilla-La Mancha. Another limitation was that, although qualitative observations were collected in the final part of the questionnaire, participants could also have been interviewed to gain a deeper understanding of their knowledge and perceptions. Future studies should try to expand this study to other Spanish regions through collaboration with the Ministry of Education and Training, highlighting the lack of research addressing school inclusion for students with rare diseases. Future studies should also include interviews with participants in their methodological designs to gain a more thorough understanding of any results.

Education systems must strive to overcome the discrimination in educational processes. It is clear from this study that there is still a long way to go in providing a proper educational response to these students with rare diseases, and we should be asking ourselves whether schools are doing all they can to offer quality, inclusive education to these students.

## Figures and Tables

**Figure 1 ijerph-19-06057-f001:**
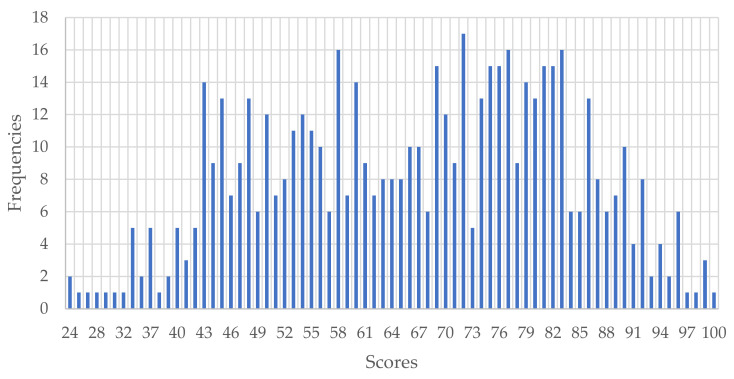
Graphical representation of the total scores for the instrument.

**Figure 2 ijerph-19-06057-f002:**
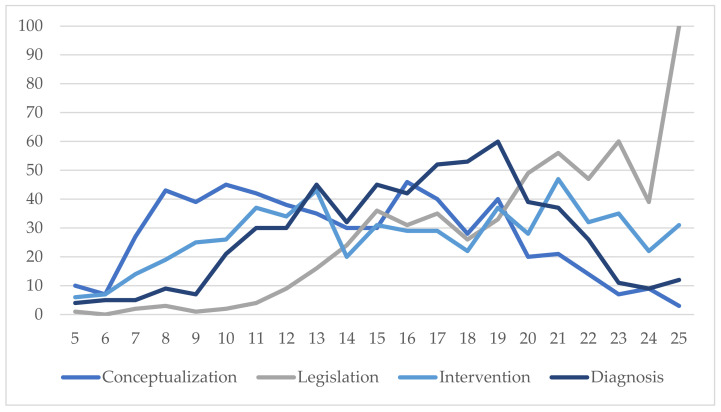
Graphical representation of the frequencies and scores for each dimension in the instrument.

**Figure 3 ijerph-19-06057-f003:**
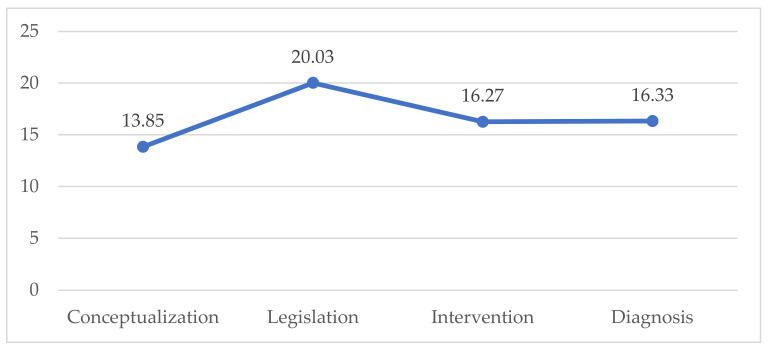
Graphical representation of the difference in means between dimensions.

**Table 1 ijerph-19-06057-t001:** Dimensions and items in the instrument.

Dimension	Items
Conceptualization	1. I know what a rare disease is 2. I know the categories that the different types of rare diseases fall into 3. I know how prevalent the rare diseases I have dealt with in my school are at a national or international level. 4. When I have had a student with a rare disease in my class, I knew what their main characteristics were 5. I have maintained contact from the school with patient advocacy groups that deal with rare diseases
Legislation	6. I know the educational inclusion legislation in our region 7. I know the content of the most recent education legislation, especially about student educational needs 8. I have read and understand the article in the Spanish Constitution which describes the right to education for all 9. I understand the importance of there being a law aimed at educational inclusion 10. Within my teaching, I believe inclusive education to be important, and therefore the principles of inclusion must be applied in the classroom
Intervention	11. I have had, or still have, a student in my class with a rare disease 12. I have educated myself about the signs and symptoms students may exhibit if they suffer from a rare disease 13. I think the family-school relationship is essential for proper intervention for students with rare diseases 14. I know specific activities to do with these students15. I am able to advise and guide other teachers about activities with these students
Diagnosis	16. I know the warning signs a student with a possible rare disease may present 17. I educate myself and try to contact other diagnosed cases in order to improve the education I give to students with rare diseases that I might encounter as a teacher 18. I know how to clinically diagnose a rare disease 19. I think the relationship between the school and the families is essential in order to be able to properly detect and help an initial diagnosis 20. I am aware of the need for more research to help diagnosis and treatment

**Table 2 ijerph-19-06057-t002:** Descriptive statistics for the instrument total score.

Minimum	Maximum	M	SD	Asymmetry	Kurtosis
24	100	66.49	16.41	−0.19	−0.82

**Table 3 ijerph-19-06057-t003:** Frequencies and percentages for items in the Conceptualization dimension.

Item	Scores									
	1		2		3		4		5	
	F	%	f	%	f	%	f	%	f	%
IT1	14	2.44	74	12.89	125	21.78	174	30.31	187	32.58
IT2	246	42.86	147	25.61	114	19.86	51	8.89	16	2.79
IT3	228	39.72	133	23.17	107	18.64	65	11.32	41	7.14
IT4	66	11.50	132	23.00	132	23.00	156	27.18	88	15.33
IT5	146	25.44	129	22.47	117	20.38	119	20.73	63	10.98

**Table 4 ijerph-19-06057-t004:** Statistics for central tendency, dispersion, and distribution for the Conceptualization dimension and its items.

Item	M	SD	Asymmetry	Kurtosis
IT1	3.78	1.11	−0.55	−0.65
IT2	2.03	1.11	0.82	−0.25
IT3	2.23	1.28	0.72	−0.62
IT4	3.12	1.25	−0.11	−1.04
IT5	2.69	1.34	0.21	−1.18
Total	13.85	4.74	0.20	−0.88

**Table 5 ijerph-19-06057-t005:** Frequencies and percentages for items in the Legislation dimension.

Item	Scores									
	1		2		3		4		5	
	F	%	f	%	f	%	f	%	f	%
IT6	34	5.92	43	7.49	118	20.56	157	27.35	222	38.68
IT7	118	20.56	80	13.94	97	16.90	134	23.34	145	25.26
IT8	18	3.14	39	6.79	95	16.55	150	26.13	272	47.39
IT9	6	1.05	9	1.57	57	9.93	135	23.52	367	63.94
IT10	8	1.39	14	2.44	51	8.89	147	25.61	354	61.67

**Table 6 ijerph-19-06057-t006:** Statistics for central tendency, dispersion, and distribution for the Legislation dimension and its items.

Item	M	SD	Asymmetry	Kurtosis
IT6	3.85	1.19	−0.84	−0.15
IT7	3.19	1.47	−0.24	−1.34
IT8	4.08	1.09	−1.06	0.31
IT9	4.48	0.82	−1.71	2.99
IT10	4.44	0.86	−1.73	3.03
Total	20.03	4.10	−0.66	−0.18

**Table 7 ijerph-19-06057-t007:** Frequencies and percentages for items in the Intervention dimension.

Item	Scores									
	1		2		3		4		5	
	f	%	f	%	f	%	f	%	f	%
IT11	191	33.28	127	22.13	82	14.29	61	10.63	113	19.69
IT12	78	13.59	79	13.76	57	9.93	124	21.60	236	41.11
IT13	10	1.74	19	3.31	39	6.79	112	19.51	394	68.64
IT14	106	18.47	125	21.78	121	21.08	135	23.52	87	15.16
IT15	201	35.02	86	14.98	111	19.34	105	18.29	71	12.37

**Table 8 ijerph-19-06057-t008:** Statistics for central tendency, dispersion, and distribution for the Intervention dimension and its items.

Item	M	SD	Asymmetry	Kurtosis
IT11	2.61	1.52	0.44	−1.28
IT12	3.63	1.47	−0.64	−1.06
IT13	4.50	0.89	−2.03	3.88
IT14	2.95	1.34	0.01	−1.20
IT15	2.58	1.43	0.30	−1.29
Total	16.27	5.40	−0.10	−1.11

**Table 9 ijerph-19-06057-t009:** Frequencies and percentages for the items in the Diagnosis dimension.

Item	Scores									
	1		2		3		4		5	
	f	%	f	%	f	%	f	%	f	%
IT16	160	27.87	146	25.44	130	22.65	107	18.64	31	5.40
IT17	106	18.47	127	22.13	121	21.08	144	25.09	76	13.24
IT18	278	48.43	126	21.95	103	17.94	45	7.84	22	3.83
IT19	24	4.18	41	7.14	53	9.23	113	19.69	343	59.76
IT20	7	1.22	9	1.57	18	3.14	68	11.85	472	82.23

**Table 10 ijerph-19-06057-t010:** Statistics for central tendency, dispersion, and distribution for the Diagnosis dimension and its items.

Item	M	SD	Asymmetry	Kurtosis
IT16	2.48	1.23	0.33	−0.99
IT17	2.93	1.32	0.00	−1.18
IT18	1.97	1.15	0.98	−0.00
IT19	4.24	1.14	−1.45	1.07
IT20	4.72	0.71	−3.18	10.99
Total	16.33	4.22	−0.26	−0.36

## Data Availability

Due to the anonymity and confidentiality of the data obtained, the authors have not reported any of the data obtained, the purpose of which is exclusively the development of this research.

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
