# Peer review of "Rare Diseases in the Educational Field: Knowledge and Perceptions of Spanish Teachers"

_ijerph, 2022, doi:10.3390/ijerph19106057_

Round 1

Reviewer 1 Report

Thank you for the opportunity to revise this paper. I think the authors deal with an interesting topic that deserves scholarly attention. In general, the paper is well organized and written and can make a relevant contribution.

A few comments that I hope the authors find interesting:

The paper starts with a definition, maybe it would be convenient to support this with a reference? (even more in case it is a literal quote)

The second paragraph introduces legislation from Spain. It would be convenient to first describe the context of the research, explaining why the focus is on Spain.

Considering there are no formal hypotheses, and that the goal of the paper as stated is “to assess what non-university teachers in one Spanish region know and think” it seems to me this type of research would benefit from qualitative analysis (interviews, focus group etc) with participants. If this is no longer possible to conduct, it should be mentioned in the limitations and future avenues of research.

The sample is skewed towards women. This might not be necessarily wrong, for example if in the total population there is also a majority of women in this positions, but it has to be explained.

Regarding the conclusions, it is recommended to improve them from the inclusion of bibliography that reinforces the aspects considered in them.

I hope the authors find my comments constructive and I wish them good luck in their project!

Author Response

Dear Reviewer,

We appreciate the review work done, thank you very much for considering our work. The following is a response to each of the reviews carried out.

Best regards

---

Reviewer. The paper starts with a definition, maybe it would be convenient to support this with a reference? (even more in case it is a literal quote)

  • It is not a literal quote. Text and reference have been included (line 25).

Reviewer. The second paragraph introduces legislation from Spain. It would be convenient to first describe the context of the research, explaining why the focus is on Spain.

  • We have added: “where the researchers work as educators and hence where this study took place” (lines 44-45).

Reviewer. Considering there are no formal hypotheses, and that the goal of the paper as stated is “to assess what non-university teachers in one Spanish region know and think” it seems to me this type of research would benefit from qualitative analysis (interviews, focus group etc) with participants. If this is no longer possible to conduct, it should be mentioned in the limitations and future avenues of research.

  • We have added “Another limitation was that, although qualitative observations were collected in the final part of the questionnaire, participants could also have been interviewed to gain a deeper understanding of their knowledge and perceptions. (lines 577-582)
  • We have added: “Future studies should also include interviews with participants in their methodological designs to gain a more thorough understanding of any results” (lines 585-587).

Reviewer. The sample is skewed towards women. This might not be necessarily wrong, for example if in the total population there is also a majority of women in this positions, but it has to be explained.

  • We have added: “this is a similar ratio of men to women who teach in schools in this region (men 29.67%, women 70.33%)” [27] (lines 153-155).

Reviewer. Regarding the conclusions, it is recommended to improve them from the inclusion of bibliography that reinforces the aspects considered in them.

  • We have revised the conclusions, including references and one more limitation of the study.

Reviewer 2 Report

This is a very interesting study that has many implications. There are a few places where the authors should consider changing their word usage:

Are “non-university teachers” those that teach students below the university level or teachers that have not taken classes at a university? Some of the wording makes it sound as if the teachers have not gone to university. In the US we refer to these teachers as K-12 (kindergarten through 12th grade) or TK-12 (transitional kindergarten). I’m wondering if what the authors mean is “teachers of students prior to university level”. If so, could that phrase be used? Is there another way to define this group of teachers instead?

What does “a qualitative analysis was performed by considering the teacher’s contributions in the observations section of the instrument.” mean on line 209? This appears to be the only place where observations are mentioned. If they were performed, additional information is needed.

“One must also be aware of the high levels of suffering in families of students with rare diseases, which means it is essential to have a multidisciplinary approach from the full educational community in order to raise levels of awareness and knowledge.” Lines 284-286 – is this an over generalized statement? This makes it sound as if all families of students with rare diseases “suffer” immensely – this cannot be universally true.

“The data about when children are diagnosed with these diseases under-scores the importance of the school years [41]: 69.9% of rare diseases are diagnosed in the pediatric age group, 18.2% are diagnosed in that age group and in adulthood, and 11.9% are diagnosed in adulthood.” Perhaps age ranges can also be added – especially to the middle group since it is unclear what the age characteristics of that group are. Lines 405-408

Lastly, it is standard formatting to offset long quotes at the beginning of the article with larger margins to emphasize the fact that they are quotes. Otherwise it appears as if the authors themselves wrote these paragraphs. Did the authors check the available styles for quotes in the word processor? Perhaps the journal can help clarify what should be done.

Author Response

Dear Reviewer,

We appreciate the review work done, thank you very much for considering our work. The following is a response to each of the reviews carried out.

Best regards

---

Reviewer. Are “non-university teachers” those that teach students below the university level or teachers that have not taken classes at a university? Some of the wording makes it sound as if the teachers have not gone to university. In the US we refer to these teachers as K-12 (kindergarten through 12th grade) or TK-12 (transitional kindergarten). I’m wondering if what the authors mean is “teachers of students prior to university level”. If so, could that phrase be used? Is there another way to define this group of teachers instead?

  • We have specified that the participating teachers teach in schools (lines 136-137).

Reviewer. What does “a qualitative analysis was performed by considering the teacher’s contributions in the observations section of the instrument.” mean on line 209? This appears to be the only place where observations are mentioned. If they were performed, additional information is needed.

  • We have added: “This information made it possible to complete the results, as can be seen in the Results section below” (lines 189-190).
  • We have added: “supported by the teachers’ observations and suggestions for improvement based on the qualitative analysis” (lines 229-231).

Reviewer. “One must also be aware of the high levels of suffering in families of students with rare diseases, which means it is essential to have a multidisciplinary approach from the full educational community in order to raise levels of awareness and knowledge.” Lines 284-286 – is this an over generalized statement? This makes it sound as if all families of students with rare diseases “suffer” immensely – this cannot be universally true.

  • We have added: “that many families of students with rare diseases experience” (lines 309-310).

Reviewer. “The data about when children are diagnosed with these diseases under-scores the importance of the school years [41]: 69.9% of rare diseases are diagnosed in the pediatric age group, 18.2% are diagnosed in that age group and in adulthood, and 11.9% are diagnosed in adulthood.” Perhaps age ranges can also be added – especially to the middle group since it is unclear what the age characteristics of that group are. Lines 405-408

  • We have modified the text to state: “69.9% of rare diseases are diagnosed in the pediatric age group (up to 18 years old, depending on the region), 18.2% are diagnosed either in that age group or in adulthood, and 11.9% are diagnosed in adulthood (from the age of 18 onwards)” (lines 432-435).

Reviewer. Lastly, it is standard formatting to offset long quotes at the beginning of the article with larger margins to emphasize the fact that they are quotes. Otherwise it appears as if the authors themselves wrote these paragraphs. Did the authors check the available styles for quotes in the word processor? Perhaps the journal can help clarify what should be done.

  • We have increased the margins on these quotes (lines 38-44, 48-54 and 104-108).

Reviewer 3 Report

I read the paper entitled "Rare Diseases in the Educational Field: Teachers' knowledge and perceptions". This is a very interesting and original study. I suggest it be published after the authors respond to my comments below:

Title:

The authors should mention in the title that the study concerns Spanish teachers.

Materials and Methods:

- It surprises me that the researchers do not mention the years of professional experience of the sample and its age. If data are available, I would ask the authors to add it to better describe the sample. If they do not exist, they should mention it in the limitations of this study.

-If the data of professional experience are available, I would ask the authors to at least consider whether there is a correlation with the values of the questionnaire responses.

Results and Discussion

-Please separate the Results from the Discussion.

-In the Results I would ask the authors to provide information on whether the answers to the questionnaire were statistically different between men and women.

- Line 481: "It is clear from the chart that the Legislation dimension had the highest scores." The finding should also be supported statistically.

- I would suggest that the authors transfer the tables with Frequencies and percentages for items to an appendix.

Conclusions:

The Conclusions take up an unusually large amount of space in the manuscript and are more like a Discussion. Please limit the conclusions to one or two paragraphs.

Author Response

Dear Reviewer,

We appreciate the review work done, thank you very much for considering our work. The following is a response to each of the reviews carried out.

Best regards

---

Reviewer. The authors should mention in the title that the study concerns Spanish teachers.

  • We have changed the title to: “Knowledge and perceptions of Spanish teachers” (lines 2-3).

Reviewer. It surprises me that the researchers do not mention the years of professional experience of the sample and its age. If data are available, I would ask the authors to add it to better describe the sample. If they do not exist, they should mention it in the limitations of this study. If the data of professional experience are available, I would ask the authors to at least consider whether there is a correlation with the values of the questionnaire responses.

  • We have added: “The distribution of participant ages was as follows: 15 (2.61%) were aged 21-25; 122 (21.26%) were aged 26-35; 296 (51.57%) were aged 36-50; and 141 (24.56%) were over 50 years old. The participants’ professional experience was as follows: 83 (14.46%) had be-tween 0-5 years’ experience; 165 (28.75%) had between 6-15 years, 210 (36.58%) had be-tween 16-25; and 116 (20.21%) had more than 26 years of teaching experience”. (lines 155-160).

Reviewer. Please separate the Results from the Discussion.

  • We have split this into two sections.

Reviewer. Line 481: "It is clear from the chart that the Legislation dimension had the highest scores." The finding should also be supported statistically.

  • We have added the means and standard deviations (lines 508-509).

Reviewer. I would suggest that the authors transfer the tables with Frequencies and percentages for items to an appendix.

  • We incorporated the tables in the text to make it easier for readers to understand the results. If the reviewer thinks it would be better to put them in an appendix, we have no problem doing so.

Reviewer. The Conclusions take up an unusually large amount of space in the manuscript and are more like a Discussion. Please limit the conclusions to one or two paragraphs.

Authors. We have split out a separate Discussion section from the Conclusions so that we could do this and added another limitation to the conclusions. (lines 585-587). The authors thank you for this review.

Reviewer 4 Report

Thank you for the opportunity to revise the paper titled: “ Rare Diseases in the Educational Field: Teachers’ knowledge 2 and perceptions”.

First of all I would like to congratulate the authors for a well written paper, dealing with an interesting and relevant topic. The paper in general seems well conducted, but I have a few comments that I hope the authors find constructive:

 First, the authors show a “structured abstract”, but I am not sure this is what the journal requests. It seems this journal usually provides “normal” abstracts in the form of a paragraph.

I would recommend removing the expression “One must not forget that one…” which seems more commonly used in other languages than in English and the sentence can read equally fine without it.

Please check small typos in the spacing in section 2.1.

The sample is skewed towards women, is this because the overall population is also more skewed towards women?

Would it be possible to provide other descriptive statistics of the sample (age, years of experience, disciplines, etc).

Only one limitation is mentioned, but I can imagine there would be many other things the authors could not ask (due to space limitations in the questionnaire or example). It would be good to reflect on this as this can help interested researchers who may want to expand this line of research, to study additional topics that arise from your study.

I wish the authors good luck!

Author Response

Dear Reviewer,

We appreciate the review work done, thank you very much for considering our work. The following is a response to each of the reviews carried out.

Best regards

---

Reviewer. First, the authors show a “structured abstract”, but I am not sure this is what the journal requests. It seems this journal usually provides “normal” abstracts in the form of a paragraph.

  • The template indicates that the abstract must be structured in the following parts: Background, Methods, Results and Conclusions. We apologize for the inconvenience and thank you for your contribution.

Reviewer. I would recommend removing the expression “One must not forget that one…” which seems more commonly used in other languages than in English and the sentence can read equally fine without it.

  • We have removed that expression (line 62).

Reviewer. Please check small typos in the spacing in section 2.1.

  • The errors in section 2.1 have been corrected, we apologize for the inconvenience.

Reviewer. The sample is skewed towards women, is this because the overall population is also more skewed towards women?

  • We have added: “this is a similar ratio of men to women who teach in schools in this region (men 29.67%, women 70.33%)” [27] (lines 153-155).

Reviewer. Would it be possible to provide other descriptive statistics of the sample (age, years of experience, disciplines, etc.).

  • We have added: “The distribution of participant ages was as follows: 15 (2.61%) were aged 21-25; 122 (21.26%) were aged 26-35; 296 (51.57%) were aged 36-50; and 141 (24.56%) were over 50 years old. The participants’ professional experience was as follows: 83 (14.46%) had be-tween 0-5 years’ experience; 165 (28.75%) had between 6-15 years, 210 (36.58%) had be-tween 16-25; and 116 (20.21%) had more than 26 years of teaching experience”. (lines 155-160).

Reviewer. Only one limitation is mentioned, but I can imagine there would be many other things the authors could not ask (due to space limitations in the questionnaire or example). It would be good to reflect on this as this can help interested researchers who may want to expand this line of research, to study additional topics that arise from your study.

  • We have added “Another limitation was that, although qualitative observations were collected in the final part of the questionnaire, participants could also have been interviewed to gain a deeper understanding of their knowledge and perceptions. (lines 577-582)
  • We have added: “Future studies should also include interviews with participants in their methodological designs to gain a more thorough understanding of any results” (lines 585-587).

Reviewer 5 Report

This study tackles on rare diseases among student population and targets specifically the level of knowledge of these disorders among non-university teachers in one Spanish region know using a questionnaire covering various dimensions for analysis, created for this purpose. I have a series of comments:

  • the random stratified sampling procedure should be explained
  • It is not clear if this is the validation study for the questionnaire employed in the study. If so, many other psychometric properties should investigated….
  • for instance, concurrent validity is a key measure
  • Combining results and discussion makes the paper verbose and dilutes the key messages. If the authors decide to keep this structure, they should at least use subheadings to facilitate readability.
  • lines 45-50: if this is a direct quote it should be italicized or with quotation marks. No need of indicating the pages in the text
  • There are typos and grammar inconsistencies that need to be revised

Author Response

Dear Reviewer,

We appreciate the review work done, thank you very much for considering our work. The following is a response to each of the reviews carried out.

Best regards

---

Reviewer. The random stratified sampling procedure should be explained.

  • The type of sampling we used was simple random, not stratified random. We apologize for the error.

Reviewer. It is not clear if this is the validation study for the questionnaire employed in the study. If so, many other psychometric properties should investigated…. for instance, concurrent validity is a key measure

  • For reasons of space and content, the entire validation process carried out for the questionnaire has not been included. Only an extract of it is included in the article, tables and graphs are not included for the reasons indicated.

Reviewer. Combining results and discussion makes the paper verbose and dilutes the key messages. If the authors decide to keep this structure, they should at least use subheadings to facilitate readability.

  • We have separated the Results and Discussion sections. We have added a specific Discussion section and added another limitation of the research to improve the Conclusion section (lines 585-587). Thank you for this comment.

Reviewer. Lines 45-50: if this is a direct quote it should be italicized or with quotation marks. No need of indicating the pages in the text

  • We have increased the margins on these quotes (lines 38-44, 48-54 and 104-108).

Reviewer. There are typos and grammar inconsistencies that need to be revised

  • The text of the article has been reviewed and corrected by a native English speaker.

Round 2

Reviewer 5 Report

I still think that details on the validation procedure should be presented as Supplementary In formation but I will leave this decision to the Editor.